# Numb provides a fail-safe mechanism for intestinal stem cell self-renewal in adult *Drosophila* midgut

**Mengjie Li[1], Aiguo Tian[1†], Jin Jiang[1,2]\***

[1]Department of Molecular Biology, University of Texas Southwestern Medical Center at Dallas, Dallas, United States; [2]Department of Pharmacology, University of Texas Southwestern Medical Center at Dallas, Dallas, United States

**\*For correspondence:**
jin.jiang@utsouthwestern.edu

**Present address:** [†]Department of Biochemistry and Molecular Biology, Tulane University School of Medicine, Louisiana Cancer Research Center, New Orleans, United States

**Competing interest:** The authors declare that no competing interests exist.

## eLife Assessment

The authors examine the role of Numb, a Notch inhibitor, in intestinal stem cell self-renewal in *Drosophila* during homeostasis and regeneration. The significance is **important** as the authors demonstrate the ISC maintenance is reduced when both BMP signaling and Numb expression is reduced. The strength of evidence is **convincing** as large sample sizes and statistical analyses are provided.

**Abstract** Stem cell self-renewal often relies on asymmetric fate determination governed by niche signals and/or cell-intrinsic factors but how these regulatory mechanisms cooperate to promote asymmetric fate decision remains poorly understood. In adult *Drosophila* midgut, asymmetric Notch (N) signaling inhibits intestinal stem cell (ISC) self-renewal by promoting ISC differentiation into enteroblast (EB). We have previously shown that epithelium-derived Bone Morphogenetic Protein (BMP) promotes ISC self-renewal by antagonizing N pathway activity (Tian and Jiang, 2014). Here, we show that loss of BMP signaling results in ectopic N pathway activity even when the N ligand Delta (Dl) is depleted, and that the N inhibitor Numb acts in parallel with BMP signaling to ensure a robust ISC self-renewal program. Although Numb is asymmetrically segregated in about 80% of dividing ISCs, its activity is largely dispensable for ISC fate determination under normal homeostasis. However, Numb becomes crucial for ISC self-renewal when BMP signaling is compromised. Whereas neither Mad RNA interference nor its hypomorphic mutation led to ISC loss, inactivation of Numb in these backgrounds resulted in stem cell loss due to precocious ISC-to-EB differentiation. Furthermore, we find that *numb* mutations resulted in stem cell loss during midgut regeneration in response to epithelial damage that causes fluctuation in BMP pathway activity, suggesting that the asymmetrical segregation of Numb into the future ISC may provide a fail-save mechanism for ISC self-renewal by offsetting BMP pathway fluctuation, which is important for ISC maintenance in regenerative guts.

## Introduction

Adult organs such as intestine rely on resident stem cells to replenish damaged tissues during homeostasis and regeneration (*Biteau et al., 2011*; *Jiang and Edgar, 2012*). *Drosophila* midgut, an equivalent of mammalian small intestine, has emerged as a powerful system to study stem cell biology in adult tissue homeostasis and regeneration (*Casali and Batlle, 2009*; *Jiang and Edgar, 2011*; *Jiang et al., 2016*). Intestinal stem cells (ISCs) in adult midguts are localized at the basal side of the gut epithelium where they can undergo asymmetric cell division to produce renewed ISCs and enteroblasts (EBs) that differentiate into enterocytes (ECs) (*Micchelli and Perrimon, 2006*; *Ohlstein and Spradling, 2006*).

At low frequency, an ISC daughter is fated to preEE/EEP that differentiates into two enteroendocrine (EE) cells after another round of cell division (*Biteau and Jasper, 2014*; *Beehler-Evans and Micchelli, 2015*; *Zeng and Hou, 2015*; *Chen et al., 2018*). About 20% ISCs undergo symmetric cell division to produce two ISCs or two EBs (*O'Brien et al., 2011*; *Goulas et al., 2012*; *Tian and Jiang, 2014*). The decision between ISC self-renewal and differentiation into EB lineage is controlled by Notch (N) signaling whereby N activation drives ISC differentiation into EB (*Micchelli and Perrimon, 2006*; *Ohlstein and Spradling, 2006*; *Ohlstein and Spradling, 2007*; *Bardin et al., 2010*). The asymmetric N signaling between ISC and EB is influenced by Par/integrins-directed asymmetric cell division and differential BMP signaling (*Goulas et al., 2012*; *Tian and Jiang, 2014*). EC-produced BMP ligands containing Decapentaplegic (Dpp) and Glass bottom boat (Gbb) heterodimers are secreted basally and concentrated on the basement membrane aligning the basal side of the gut epithelium (*Tian and Jiang, 2014*). After asymmetric cell division of ISCs, basally localized daughter cells transduce higher levels of BMP signaling activity than the apically localized daughter cells and the differential BMP signaling promotes ISC self-renewal by antagonizing N pathway activity through an unknown mechanism (*Tian and Jiang, 2014*; *Tian and Jiang, 2017*).

The N pathway inhibitor Numb plays a decisive role in asymmetric cell fate determination in *Drosophila* peripheral and central nervous systems whereby Numb segregates asymmetrically into one daughter during division of a neuronal precursor cell and confers distinct fates to the two daughter cells (*Uemura et al., 1989*; *Rhyu et al., 1994*; *Spana et al., 1995*). The mammalian homologs of Numb are also critical for asymmetric fate determination during neurogenesis and myogenesis (*Zhong et al., 1996*; *Conboy and Rando, 2002*; *Petersen et al., 2002*; *Shen et al., 2002*; *Petersen et al., 2004*). Previous studies showed that, during asymmetric division of an ISC in *Drosophila* adult midgut, Numb was preferentially segregated into the basally localized daughter that becomes future ISC (*Goulas et al., 2012*; *Sallé et al., 2017*); however, many *numb* mutant clones retained ISC after many rounds of cell division although they failed to produce EE (*Bardin et al., 2010*; *Sallé et al., 2017*). Given the well-established role of Numb in blocking N pathway activity and the observation that Numb is asymmetrically segregated into the future ISC, it is surprising and puzzling that loss of Numb does not lead to ectopic N pathway activation that drives ISC-to-EB differentiation. We speculate that BMP signaling may play a more dominant role in ISC self-renewal than Numb and that the BMP signaling gradient could generate differential N signaling between the apical and basal pair of ISC daughters to generate ISC/EB binary fates even when Numb is depleted (*Tian and Jiang, 2017*). Accordingly, attenuating BMP signaling may unmask the role of Numb in ISC self-renewal.

To test this hypothesis, we employed RNA interference (RNAi) and genetic mutations to inactivate Numb in otherwise wild-type background or in midguts defective in BMP signaling due to RNAi or genetic mutation of *mothers against decapentaplegic* (*mad*), which encodes a signal transducer in BMP signaling pathway (*Sekelsky et al., 1995*; *Newfeld et al., 1997*). Consistent with our previous findings (*Tian and Jiang, 2014*), neither *mad* RNAi nor its hypomorphic mutation led to ISC loss. However, inactivation of Numb in these backgrounds resulted in stem cell loss due to precocious ISC-to-EB differentiation. By carefully examining mutant clones for multiple *numb* alleles, we also observed an increased number of *numb* clones that lack ISCs compared with wild-type control clones. Interestingly, the stem cell loss phenotype was exacerbated by feeding flies with bleomycin, which resulted in EC damage and fluctuation of BMP signaling (*Amcheslavsky et al., 2009*; *Tian et al., 2017*), underscoring an important role of Numb in ISC maintenance during gut regeneration.

## Results

### Loss of BMP signaling results in ectopic N pathway activity even when Dl is depleted

Our previous study showed that depleting the type II receptor Punt (Put) for BMP in progenitor cells (ISC/EB) resulted in precocious ISC-to-EB differentiation, leading to stem cell loss (*Tian and Jiang, 2014*). In Put deficient progenitor cells, the N pathway was activated in the absence of detectable N ligand Delta (Dl). Progenitor cells deficient for both Put and N failed to differentiate to EBs and formed stem cell-like tumors (*Tian and Jiang, 2014*). These observations imply that inactivation of Put may unleash a ligand-independent N pathway activity that drives precocious ISC-to-EB differentiation,

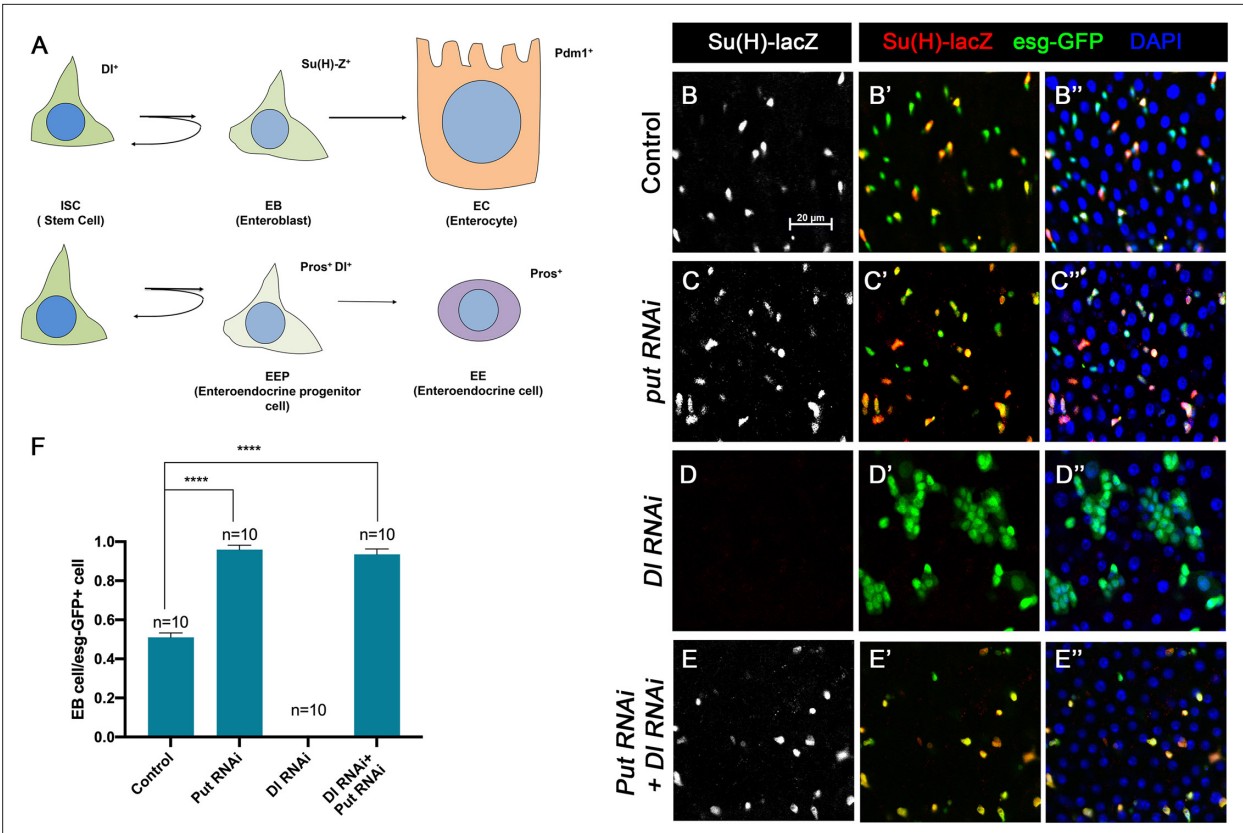

**Figure 1.** BMP signaling inhibits Dl-independent N pathway activity to promote intestinal stem cell (ISC) self-renewal. (**A**) A scheme for the ISC lineage in *Drosophila* midgut. (**B-E"**) Representative images of Control guts (**B–B"**), midguts expressing *UAS-Put-RNAi* (**C–C"**), *UAS-Dl-RNAi* (**D–D"**), or *UAS-Put-RNAi+UAS-Dl-RNAi* (**E–E"**) with *esg-Gal4ts, UAS-GFP* at 29°C for 10 days and immunostained for Su(H)-lacZ (gray or red) and GFP (green). Su(H)-lacZ is used as a marker for enteroblast (EB). DAPI (blue) staining indicates nuclei. Compared with control guts (**B–B"**), Put knockdown (**C–C"**) in precursor cells (green) caused an increase of EB pairs. Dl knockdown induced stem cell-like tumor. Put and Dl double knockdown induced a dramatic increase of EBs. (**F**) Quantification of percentage of EB cells of each genotype. Data are mean ± SD from three independent experiments. *p<0.05, ****p < 0.0001. One-way ANOVA was performed for statistical comparisons. Scale bar (20 µm) is shown in B.

The online version of this article includes the following source data for figure 1:

**Source data 1.** Source data for the quantification in *Figure 1*.

leading to stem cell depletion. However, it remains possible that a trace amount of Dl that is beyond the detection by immunostaining might activate N in Put deficient progenitor cells.

To further explore the relationship between the BMP and N pathways, we carried out genetic epistatic experiments to determine the functional relationship between Put and Dl. We depleted Put and Dl either individually or in combination via RNAi in midgut progenitor cells using the *esg-Gal4 tub-Gal80ts* (*esgts*) system, in which Gal4 is under the control of a temperature sensitive Gal80 (*McGuire et al., 2004*). *UAS-GFP* was included in the *esgts* system to mark all precursor cells whereas *Su(H)-lacZ* (also called *Su(H)GBE-lacZ*), a transcriptional reporter of N signaling, was used to monitor N pathway activity and mark the EBs. Three- to five-day-old adult females expressing *UAS-Put-RNAi* and *UAS-Dl-RNAi* individually or in combination via *esgts* were shifted to 29°C for 10 days prior to dissection, followed by immunostained with the corresponding antibodies. For quantification, we focused on the posterior region (R4) of the midguts to minimize the variation (*Buchon et al., 2013*). In control guts, most pairs of *esg>GFP +* precursor cells, which may represent two recently divided ISC daughters, contained only one *Su(H)-lacZ* positive cell, indicating that these ISCs divided asymmetrically to produce one ISC and one EB (*Figure 1A, B–B", F*). In line with our previous findings, most *esg>GFP +* precursor pairs from the Put RNAi guts expressed *Su(H)-lacZ* in both ISC daughters, suggesting that ISCs underwent symmetric cell division to produce two EBs when Put was depleted (*Figure 1C–C", F*). By contrast, *esg>GFP+* cells in Dl RNAi guts formed ISC-like tumor clusters that were negative for

*Su(H)-lacZ* (*Figure 1D–D", F*), similar to the ISC-like tumor clusters in guts containing *Dl* or *N* mutant clones (*Micchelli and Perrimon, 2006*; *Ohlstein and Spradling, 2006*; *Ohlstein and Spradling, 2007*; *Siudeja et al., 2015*), suggesting that in Dl RNAi guts, N pathway activity was diminished. Put and Dl double RNAi suppressed the formation of ISC-like clusters (*Figure 1E*). In these guts, *esg-GFP⁺* cells exhibited *Su(H)-lacZ* expression (*Figure 1E–E", F*), suggesting that they activate the N pathway and differentiate into EBs. Hence, loss of BMP signaling resulted in ectopic N pathway activity that drives ISC-to-EB differentiation even when Dl was depleted.

## Numb is important for ISC maintenance when BMP pathway activity is attenuated

Our previous study showed that partial loss of BMP pathway activity in several genetic backgrounds including Mad RNAi and *mad¹⁻²*, a hypomorphic allele of *mad*, did not lead to ISC loss whereas more complete loss of BMP signaling in Put RNAi guts or *put* mutant ISC-lineage clones resulted in ISC loss. It is possible that a backup mechanism for ISC self-renewal may exist, which could compensate for the partial loss of BMP signaling to prevent ectopic N pathway activation that drives differentiation. During an asymmetric ISC division, the N inhibitor Numb is segregated into the basally localized daughter that becomes the future ISC (*Goulas et al., 2012*; *Sallé et al., 2017*). We hypothesized that the asymmetric distribution of Numb may provide such a backup mechanism to ensure that the basally localized ISC daughter has lower N pathway activity than the apically localized one when differential BMP signaling is compromised so that the differential N signaling between the apical and basal is still sufficient to drive asymmetric fate determination. To test this hypothesis, we inactivated Numb and Mad either individually or in combination using two independent approaches: (1) RNAi and (2) genetic mutations. For the RNAi experiments, 3- to 5-day-old females expressing *UAS-Numb-RNAi*, *UAS-Mad-RNAi*, or *UAS-Numb-RNAi + UAS-Mad-RNAi* under the control of *esgᵗˢ* were transferred to 29°C for 14 days. The guts were then dissected out for immunostaining to detect the expression of *esg>GFP*, Dl-lacZ (ISC marker), E(spl)mβ-CD2 (EB marker) and Pros (EE marker). Because preEE expressed both Dl-lacZ and Pros and Dl-lacZ signals could be found in some EBs due to its perdurance, we counted Dl-lacZ⁺ mβ-CD2⁻ Pros⁻ cells as ISCs and mβ-CD2⁺ cells as EBs. Compared with control guts (*Figure 2A–A"*), Mad (*Figure 2B–B"*), or Numb (*Figure 2C–C"*) single RNAi guts contained comparable number of Dl-lacZ⁺ mβ-CD2⁻ Pros⁻ cells and E(spl)mβ-CD2⁺ cells (*Figure 2E–G*). By contrast, in Numb and Mad double RNAi guts (*Figure 2D–D"*), there was a significant decrease in the number of precursor cells (*Figure 2E*) and Dl-lacZ⁺ mβ-CD2⁻ Pros⁻ cells (*Figure 2F*), and a simultaneous increase in the number of E(spl)mβ-CD2⁺ cells (*Figure 2G*), suggesting that inactivation of both Mad and Numb results in stem cell loss, likely due to precocious ISC-to-EB differentiation.

In the second approach, we generated guts that carried *mad¹⁻²* or *numb⁴* single mutant clones or *mad¹⁻²*, *numb⁴* double mutant clones using the MARCM system that positively mark the clones with GFP expression. Three- to five-day-old females of appropriate genotypes were heat-shocked for 1 hr for clonal induction and kept at 18°C for 14 days prior to dissection. ISCs were identified as Dl⁺ cells or mβ-CD2⁻ Pros⁻ cells containing small nuclei. ISC-containing clones (ISC⁺) and clones without ISCs (ISC⁻) were quantified for each genotype. We also quantified the size of ISC-lineage clones for each genotype by counting GFP⁺ cells in individual clones. Consistent with previous findings (*Tian and Jiang, 2014*; *Sallé et al., 2017*), the average size of *mad¹⁻²* clones is significantly larger than the control clones (*Figure 3A–A', B–B', E–E', F–F', K*) whereas *numb⁴* clones had similar clone size distribution compared with control clones (*Figure 3C–C', G–G', K*). In addition, most of *mad¹⁻²* or *numb⁴* clones contained at least one ISC similar to control clones (*Figure 3K*). However, the average size of *mad¹⁻² numb⁴* clones is significantly smaller than that of control clones (*Figure 3D–D', H–H', K*). More importantly, a much larger fraction of *mad¹⁻² numb⁴* clones (~40%; *n* = 252) did not contain ISC (*Figure 3L*), many of which only contained ECs with large nuclei and stained positive for Pdm1 (*Figure 3I–J'*). Taken together, these results suggest *mad¹⁻² numb⁴* double mutation leads to ISC loss.

## Inactivation of Numb and Mad leads to precocious ISC-to-EB differentiation

We employed a two-color lineage tracing system called RGT (*Tian and Jiang, 2014*; *Tian et al., 2017*) to determine whether simultaneous inactivation of Numb and Mad would change the outcome of an ISC division. In this system, FLP/FRT-mediated mitotic recombination in individual dividing ISCs

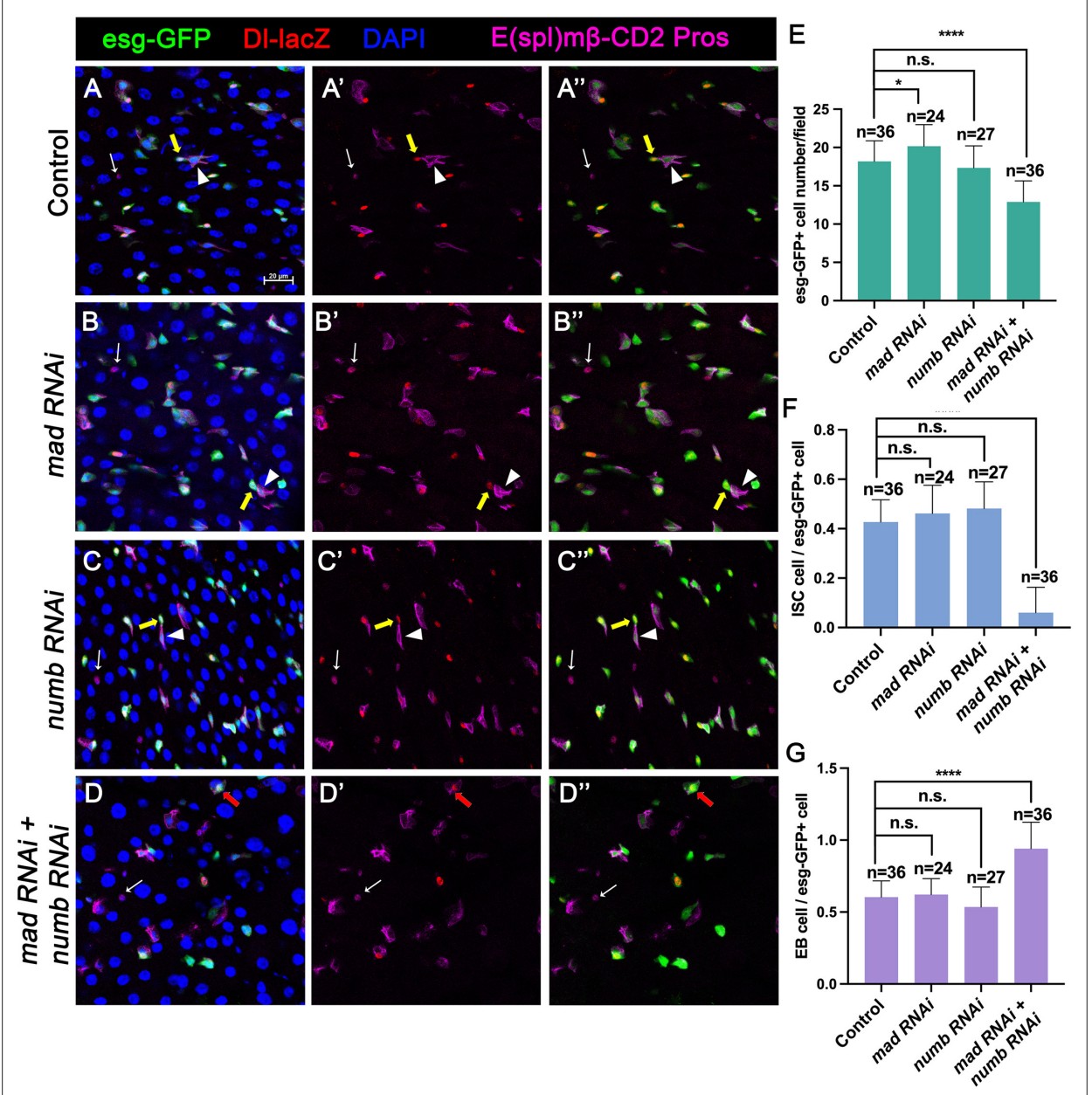

**Figure 2.** Numb is important for intestinal stem cell (ISC) maintenance when BMP pathway activity is attenuated. (**A-D"**) Representative images of adult midguts expressing *UAS-mCherry-RNAi* (Control) (**A-A"**), *UAS-Mad-RNAi* (**B-B"**), *UAS-Numb-RNAi* (**C-C"**), and *UAS-Mad-RNAi+UAS-Numb-RNAi* (**D-D"**) with *esg-Gal4ts*, *UAS-GFP* at 30°C for 14 days and immunostained for Dl-lacZ (red), E(spl)mβ-CD2 (cytoplasmic magenta), and Pros (nuclear magenta), which are markers for ISC, enteroblast (EB), and enteroendocrine (EE), respectively. DAPI (blue) staining indicates nuclei. Yellow arrows indicate ISCs (Dl-lacZ+ E(spl)mβ-CD2− Pros−), white arrowheads indicate EBs (E(spl)mβ-CD2+), and white arrows indicate EEs (Pros+) in Control, Mad, or Numb single knockdown guts. Red arrow indicated a Dl-lacZ+, E(spl) mβ-CD2+ cells in Mad and Numb double knockdown guts. Scale bar (20 µm) is presented in (**A**). (**E–G**) Quantification of number of precursor cells (**E**), percentage of ISC cells (**F**), and percentage of EB cells (**G**) of each genotype. Data are mean ± SD from three independent experiments. *p < 0.05, ****p < 0.0001. One-way ANOVA was performed for statistical comparisons.

The online version of this article includes the following source data for figure 2:

**Source data 1.** Source data for the quantification in *Figure 2E–G*.

will generate two distinctly labeled clones that express either RFP (red) or GFP (green) (*Figure 4A*). As shown schematically in *Figure 4B*, asymmetric ISC division (ISC/EB) will generate one clone with multiple cells and a twin spot that contains only one EC. Symmetric self-renewing division (ISC/ISC) will produce two multiple-cell clones whereas symmetric differentiation division (EB/EB) will produce

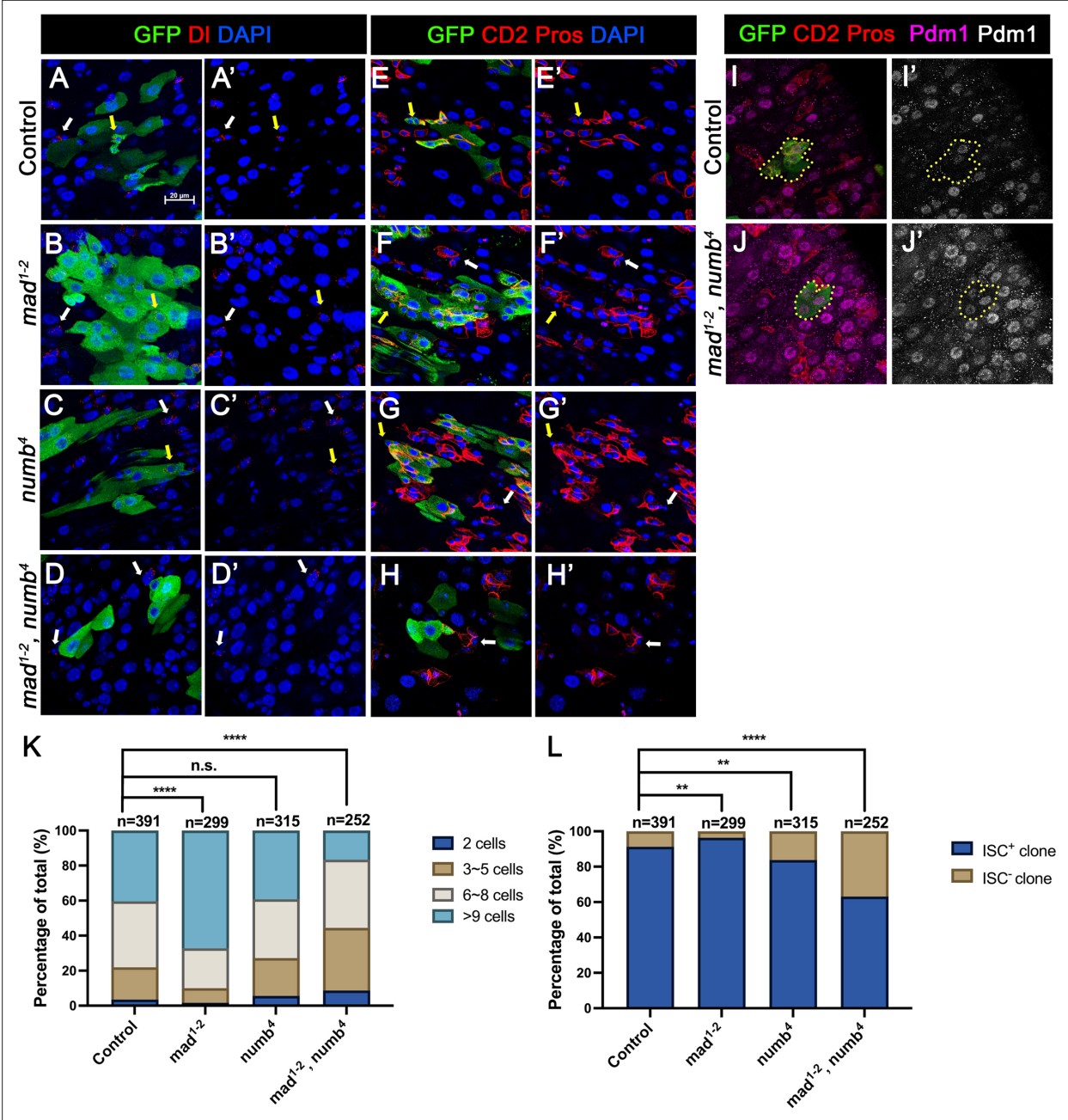

**Figure 3.** *numb* and *mad* double mutations resulted in loss of intestinal stem cell (ISC). (**A-H'**) Representative images of adult midguts containing MARCM clones (green) of *FRT40* (Control) (**A, A', E, E'**), *mad*$^{1-2}$ (**B, B', F, F'**), *numb*$^4$ (**C, C', G, G'**), and *mad*$^{1-2}$, *numb*$^4$ (**D, D', H, H'**) and immunostained for GFP (green) and Dl (red in A–D') or E(spl)mβ-CD2 and Pros (red in E–H') at 14 days (grown at 18°C) after clone induction. GFP marks the clones. DAPI (blue) staining indicates nuclei. ISCs inside and outside the clones are indicated by yellow and white arrows, respectively. (**I**) Representative images of adult midguts containing MARCM clones (green) of control (**I, I'**) or *mad*$^{1-2}$, *numb*$^4$ (**J, J'**) immunostained for GFP (green), E(spl)mβ-CD2 and Pros (red), and Pdm1 (magenta and gray). Scale bar (20 μm) is presented in (**A**). (**K**) Quantification of clone size for the indicated genotypes 14 days after clone induction. (**L**) Quantification of numbers of clones with or without ISCs. Data are mean ± SD from three independent experiments. **p < 0.01, ****p < 0.0001. $x^2$ test was performed for statistical comparisons.

The online version of this article includes the following source data for figure 3:

**Source data 1.** Source data for the quantification in *Figure 3L*.

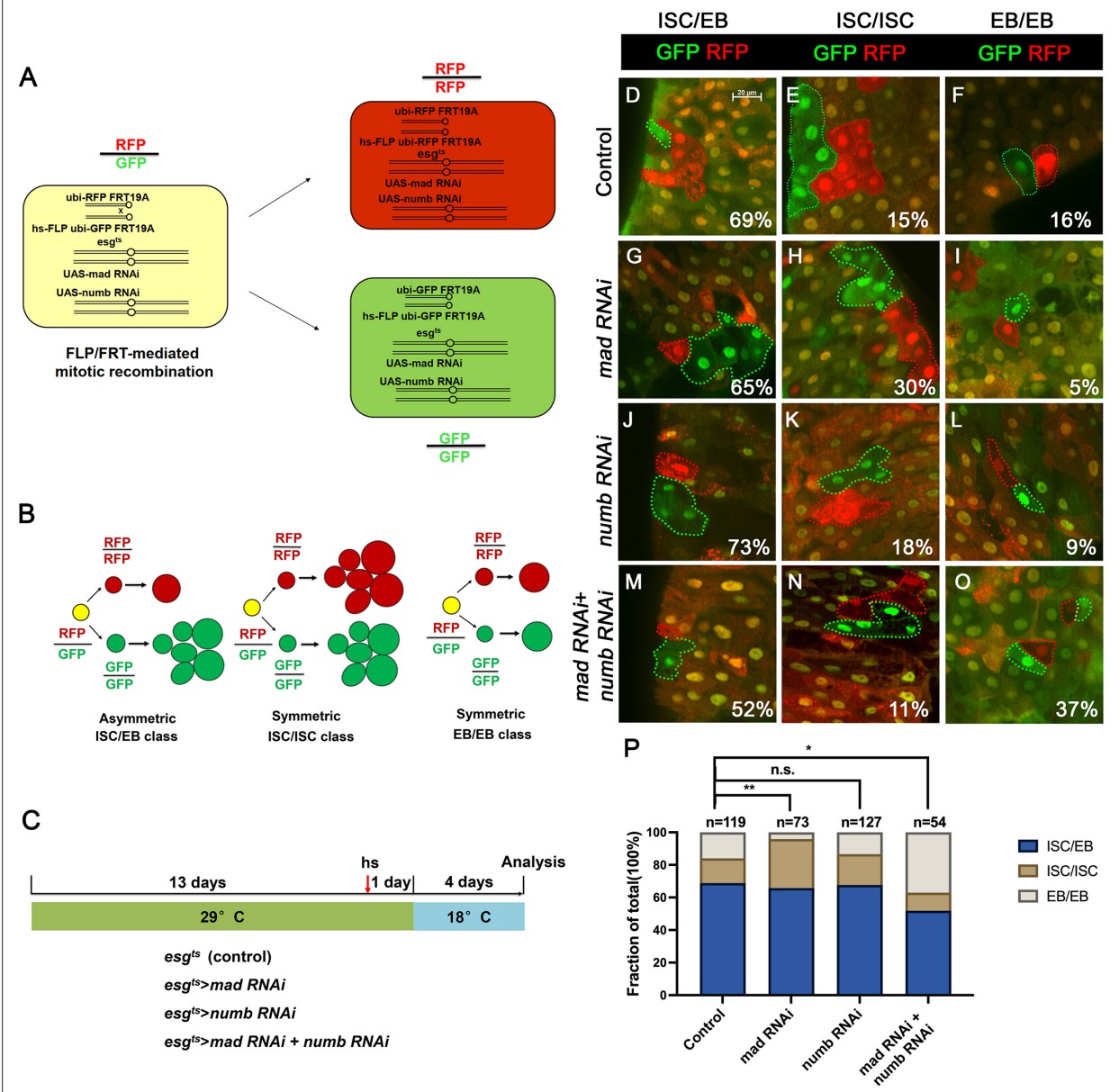

**Figure 4.** Depletion of both Numb and Mad leads to more symmetric intestinal stem cell (ISC) divisions that produce two enteroblasts (EBs). (**A**) Scheme of an ISC division that produces differentially labeled daughter cells (RFP⁺ GFP⁻ and RFP⁻ GFP⁺) through FRT-mediated mitotic recombination. Adapted from *Tian and Jiang, 2014*. (**B**) Scheme of differentially labeled twin clones generated by FLP/FRT-mediated mitotic recombination of dividing ISCs. Adapted from *Tian and Jiang, 2014*. (**C**) Scheme of twin-spot experiments. Three- to five-day-old adult flies of indicated genotype are grown at 29°C for 14 days before heat shock to induce clones. After 1-day recovery at 29°C, the flies are raised at 18°C for 4 days prior to analysis. (**D–O**) Representative images of twin-spot clones from adult midguts of the indicated genotypes. Scale bar 20 µm is shown in (**D**). (**P**) Quantification of twin spots of different classes from guts of the indicated genotypes. Data are mean ± SD from three independent experiments. *p < 0.05, **p < 0.01. $x^2$ test was performed for statistical comparisons.

The online version of this article includes the following source data for figure 4:

**Source data 1.** Source data for the quantification in *Figure 4P*.

two clones each of which contains one EC. Control or RNAi expressing adult flies containing *hs-FLP FRT19A ubi-GFPnls/FRT19A ubi-mRFPnls; esg^ts* were grown at 29°C for 8 days (for *Mad-RNAi* only) or 14 days (for control, *Numb-RNAi*, and *Numb-RNAi + Mad-RNAi*) before clone induction by heat shock at 37°C for 1 hr. After clone induction, the flies were incubated at 18°C for another 4 days before guts were dissected out for analysis (*Figure 4C*). The frequencies of ISC/EB, ISC/ISC, and EB/EB divisions

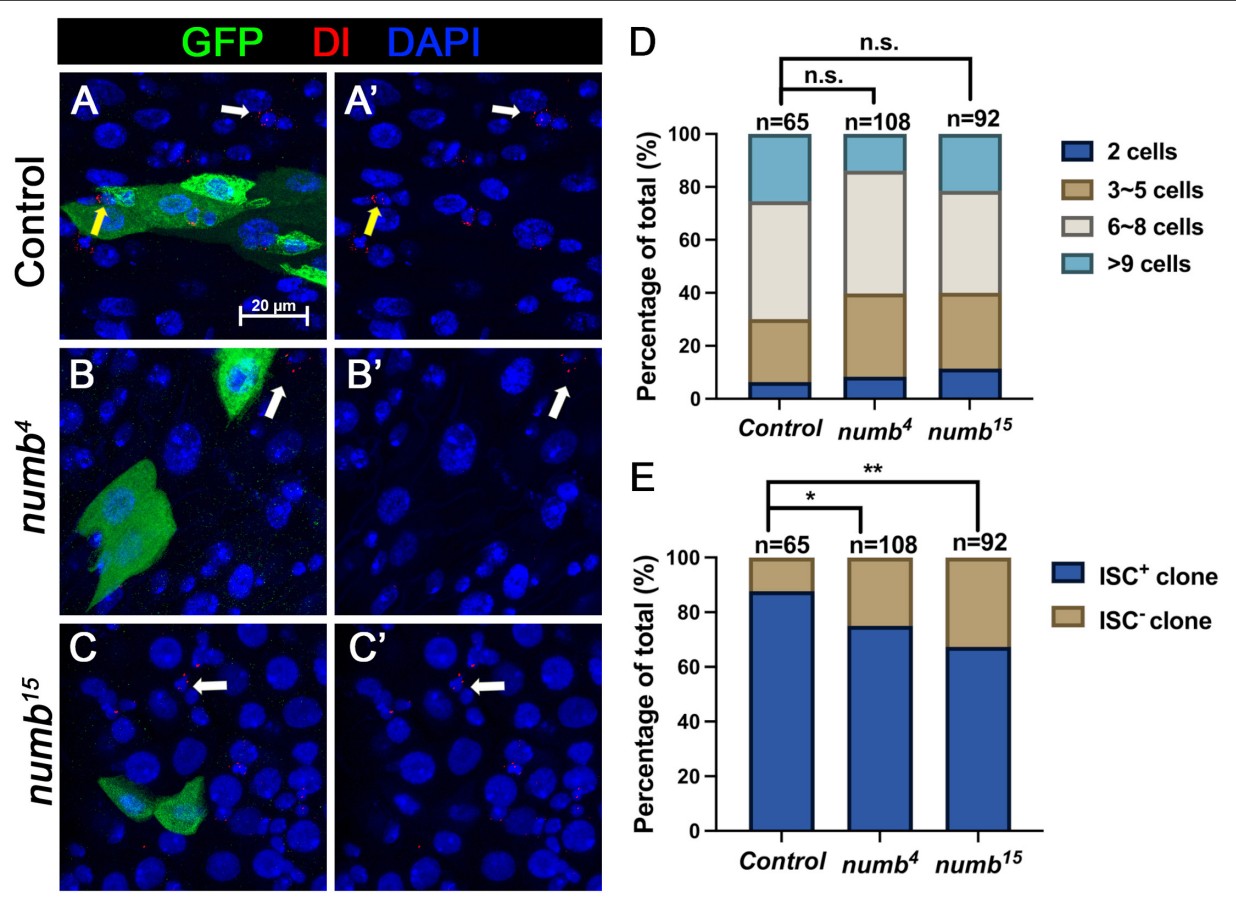

**Figure 5.** *numb* mutant clones exhibit weak stem cell loss phenotype. (**A-C'**) Representative images of adult midguts containing MARCM clone (green) of *FRT40* (Control) (**A, A'**), *numb⁴* (**B, B'**), and *numb¹⁵* (**C, C'**) and immunostained for Dl (red), GFP (green), and DAPI (blue) at 14 days after clone induction. GFP marks the clones. Intestinal stem cells (ISCs) inside and outside the clones are indicated by yellow and white arrows, respectively. Scale bar (20 µm) is shown in (**A**). (**D**) Quantification of clone size distribution for the indicated genotypes at 14 days after clone induction. (**E**) Quantification of numbers of clones with or without ISC. Data are mean ± SD from three independent experiments. *p < 0.05, **p < 0.01. $x^2$ test was performed for statistical comparisons.

The online version of this article includes the following source data and figure supplement(s) for figure 5:

**Source data 1.** Source data for the quantification in *Figure 5E*.

**Figure supplement 1.** Numb is required for enteroendocrine (EE) fate determination.

in control guts were 69%, 15%, and 16%, respectively (*n* = 119) (*Figure 4D–F, P*). Mad RNAi guts had higher frequency of ISC/ISC (30%), and lower frequency of EB/EB (5%) division compared to control guts (*n* = 73) (*Figure 4G–I, P*). The increase in symmetric self-renewing division in Mad RNAi guts is likely due to an increase in BMP ligand production in these guts because BMP signaling in EC inhibits BMP ligand expression (*Tian et al., 2017*). The frequencies of different ISC division classes in Numb RNAi guts are comparable to those of control guts (ISC/EB: 73%, ISC/ISC: 18%, EB/EB: 9%, *n* = 127) (*Figure 4J–L, P*). By contrast, Mad and Numb double RNAi guts had lower frequency of ISC/ISC division (11%) and much higher EB/EB division (37%) than control guts (*n* = 54) (*Figure 4M–P*). Thus, inactivation of Numb in backgrounds where BMP signaling was compromised altered the ISC division outcome that favors symmetric differentiation division leading to ISC loss.

## Numb mutant clones exhibit weak ISC loss phenotype

When we examined adult midguts containing *numb⁴* clones, we noticed a slight increase in the frequency of ISC⁻ clones compared to the control guts even though the average clone size of *numb⁴* clones was comparable to that of the control clones (*Figure 3K, L*), suggesting that *numb* mutation may result in a mild stem cell loss phenotype. To verify this result, we examined another *numb* allele,

numb[15]. By immunostaining for Dl expression that marks ISC, we found that both *numb[4]* and *numb[15]* clones contained similarly higher frequencies of Dl⁻ clones than the control clones (**Figure 5A–C', E**). Consistent with previous findings (**Bardin et al., 2010**; **Sallé et al., 2017**), most *numb[15]* clones grew into large size similar to the control clones (**Figure 5D**), suggesting that many ISC⁻ *numb* clones lost ISC at late stages during their clonal growth. We also examined Pros expression and found that *numb* mutant clones did not contain Pros⁺ cells (**Figure 5—figure supplement 1A–C'**), which is consistent with a previous study showing that *numb* is required for EE fate regulation (**Sallé et al., 2017**).

## Numb is important for ISC maintenance during regeneration

The weak ISC loss phenotype associated with *numb* mutant clones could be due to fluctuation in BMP pathway activity under normal homeostasis because the expression of two BMP ligands Dpp and Gbb is uneven in homeostatic guts (**Tian and Jiang, 2014**). If so, the stem cell loss phenotype caused by *numb* mutations could be enhanced under conditions where tissue damage causes more dramatic and widespread fluctuation in BMP signaling activity in regenerative guts. To test this possibility, we fed adult female flies carrying either control or *numb* clones in the guts with sucrose (mock), bleomycin, or dextran sodium sulfate (DSS). In mock-treated control guts, approximately 12% ($n$ = 178) of the clones did not contain Dl⁺ cell, while 21% ($n$ = 216) of the *numb[4]* and 24% ($n$ = 219) of the *numb[15]* clones were void of stem cells (**Figure 6A–A', D–D', G–G', J**). Previous studies showed that bleomycin treatment caused EC damage and enhanced the fluctuation in BMP ligand production whereas DSS affected basement membrane organization but did not increase the fluctuation in BMP ligand production (**Amcheslavsky et al., 2009**; **Tian et al., 2017**). In guts treated with bleomycin, 12% ($n$ = 160) of control clones did not contain Dl⁺ cells (**Figure 6B–B', J**). However, bleomycin feeding resulted in a dramatic increase of Dl⁻ ISC-lineage clones in guts containing *numb* mutant clones as Dl⁺ cells were absent in 43% ($n$ = 149) of *numb[4]* and 45% ($n$ = 213) of *numb[15]* clones (**Figure 6E–E', H–H', J**). By contrast, DSS feeding did not increase the frequency of Dl⁻ clones in guts containing *numb* mutant clones, as the frequencies of Dl⁻ clones in control, *numb[4]* and *numb[15]* clonal guts were 10% ($n$ = 167), 24% ($n$ = 165), and 21% ($n$ = 141), respectively (**Figure 6C–C', F–F', I–I' J**). Bleomycin also resulted in a reduction in *numb* mutant clone size, as compared with the mock treatment (**Figure 6K**). Taken together, these results suggest that Numb plays an important role in ISC maintenance in regenerative guts in response to bleomycin-induced tissue damage.

## Discussion

Despite the prominent role of Numb in asymmetric cell fate decision in the nervous system and the observation that Numb is asymmetrically segregated in dividing ISCs, whether Numb plays any role in ISC fate determination has remained a mystery. Here, we demonstrated that Numb is important for ISC self-renewal during regeneration. We found that Numb is largely dispensable for homeostatic ISC self-renewal due to the predominant role of BMP signaling in this context. Indeed, we found that Numb becomes critical for ISC self-renewal under conditions where BMP signaling is compromised.

Previous studies did not score a stem cell loss phenotype associated with *numb* mutant clones because the majority of *numb* mutant ISC-lineage clones could grow into large size comparable to control clones (**Figures 3 and 5**; **Bardin et al., 2010**; **Goulas et al., 2012**; **Sallé et al., 2017**). Instead, Sallé et al. showed that *numb* mutant clones lacked EEs, suggesting that Numb is important for EE fate determination (**Sallé et al., 2017**), which we confirmed in this study (Figure S1). However, by carefully examining ISC/EB markers associated with *numb* mutant clones, we noticed an increase in the fraction of *numb* mutant clones that lack ISCs compared with the control clones (**Figures 3 and 5**). By introducing *mad* mutation (*mad[1-2]*) into the *numb* mutant background, we found that *numb[4] mad[1-2]* double mutant clones had a much higher frequency to lose ISCs than *numb[4]* clones even though *mad[1-2]* single mutant clones showed no ISC loss phenotype compared with control clones (**Figure 3**).

Our previous study showed that immediately after bleomycin treatment, there was an increase in the ISC population size due to a transient surge in BMP ligand production (**Tian et al., 2017**). However, during regeneration, BMP ligand production was downregulated due to the autoinhibition of BMP ligand expression by BMP signaling in ECs (**Tian et al., 2017**). The reduction in BMP ligand production promoted ISC-to-EB differentiation to reset the ISC population size back to the homeostatic level after regeneration (**Tian et al., 2017**). It is likely that asymmetric distribution of Numb in dividing

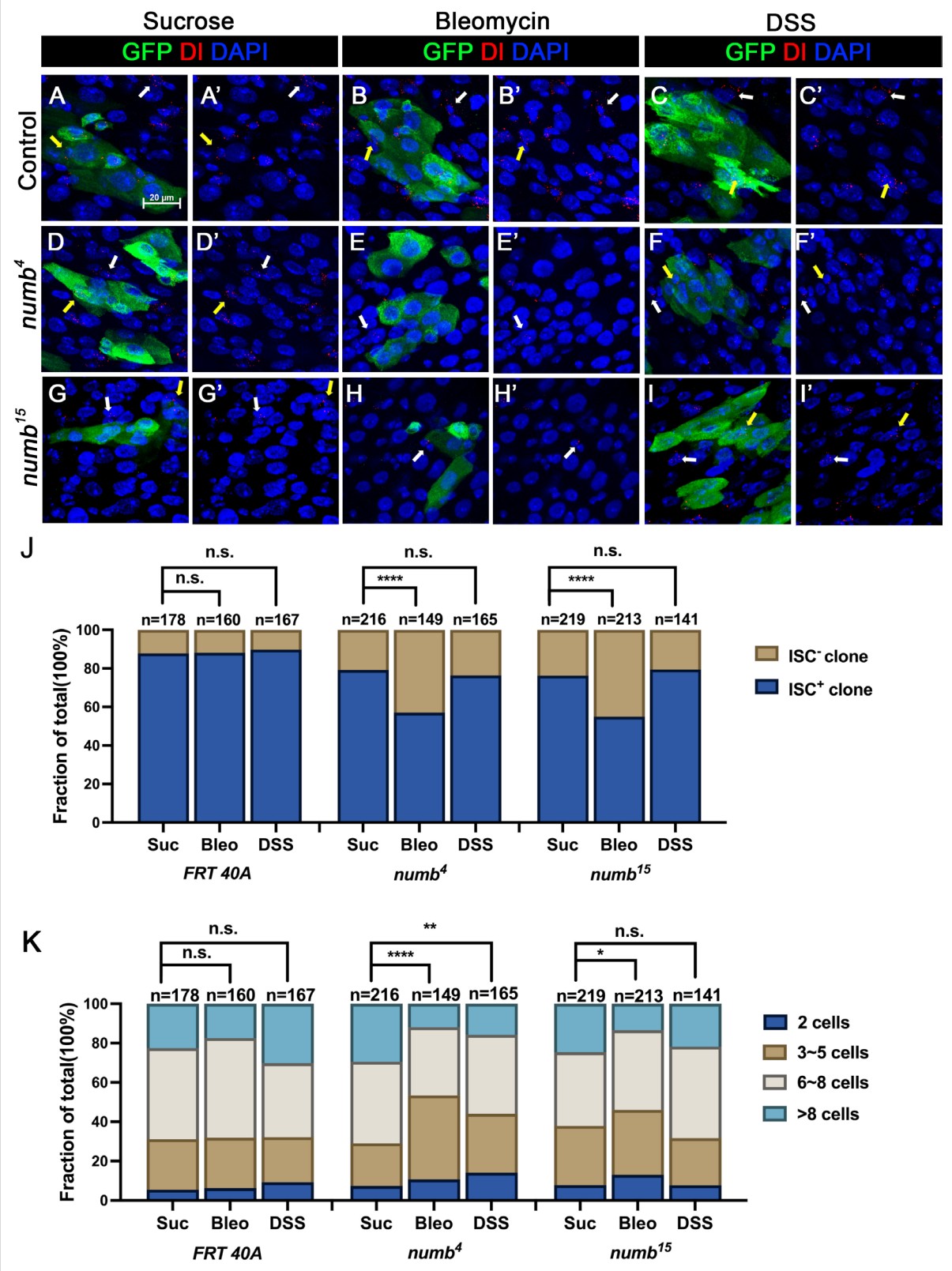

**Figure 6.** Numb is critical for intestinal stem cell (ISC) maintenance during regeneration. (**A–I'**) Adult flies of indicated genotype were treated with sucrose, bleomycin, or dextran sodium sulfate (DSS) for 24 hr at 14 days after clone induction and recovered for another 4 days before dissection. Guts containing MARCM clones of the indicated genotype were stained for GFP (green) and Dl (red). GFP marks the clones. DAPI (blue) staining indicates the nuclei. Stem cells inside and outside the clones are indicated by yellow and white arrows, respectively. Scale bar (20 μm) is shown in

*Figure 6 continued on next page*

*Figure 6 continued*

(**A**). (**J**) Quantification of the percentage of clones with or without ISCs. (**K**) Quantification of clone size distribution for the indicated genotypes. Data are mean ± SD from three independent experiments. *p < 0.05, **p < 0.01, ****p < 0.0001. $x^2$ test was performed for statistical comparisons.

The online version of this article includes the following source data for figure 6:

**Source data 1.** Source data for the quantification in *Figure 6J*.

ISCs may prevent excessive ISC-to-EB differentiation that could otherwise lead to a decreased ISC population during regeneration. Indeed, we observed an increase in the frequency of *numb* mutant clones lacking ISCs in response to bleomycin treatment, suggesting that Numb becomes critical for ISC maintenance during gut regeneration in response to tissue damage.

Based on our findings in current and previous studies, we propose the following working model to account for the cooperation between Numb and BMP signaling in the regulation of ISC self-renewal under homeostatic and tissue regeneration (*Figure 7*). Under homeostatic conditions, most ISCs divide basally so that the basally localized ISC daughters inherit Numb and transducing higher levels of BMP signaling activity than the apically situated daughter cells, the combined differential activities in BMP signaling and Numb drive robust asymmetric division outcomes to produce ISC/EB pairs (*Figure 7A*). In the *numb* mutant background, differential BMP signaling activities between the apical and basal daughter cells suffice to drive asymmetric division outcomes in most cases (*Figure 7B*). In the *mad* mutant background, the BMP signaling activity gradient becomes shallower but differential Numb activity between the apical and basal daughter cells can compensate for the compromised BMP signaling gradient to drive asymmetric division outcomes (*Figure 7C*). However, in *numb mad* double mutant background or in *numb* mutant guts damaged by bleomycin feeding, the compromised BMP signaling activity gradient alone is often insufficient to drive asymmetric division outcomes, leading to ISC loss due to symmetric EB/EB division outcomes (*Figure 7D*). One interesting question is why asymmetric Numb activity is unable to drive asymmetric ISC division outcome in the absence of BMP signaling as seen in *put* mutant background. One possibility is that the Numb level is too low in midgut ISCs so that the asymmetric Numb inheritance during ISC division is not robust enough to ensure asymmetric N signaling in the absence of BMP signaling. Indeed, previous studies indicate that endogenous Numb was undetectable using Numb antibodies (*Goulas et al., 2012*; *Couturier et al., 2013*; *Sallé et al., 2017*). Another non-mutually exclusive possibility is that Numb may not be able to counter the ligand-independent N pathway activity unleashed in *put* mutant backgrounds. Future study is needed to test these possibilities and to determine the precise mechanism by which BMP signaling inhibits N pathway activity.

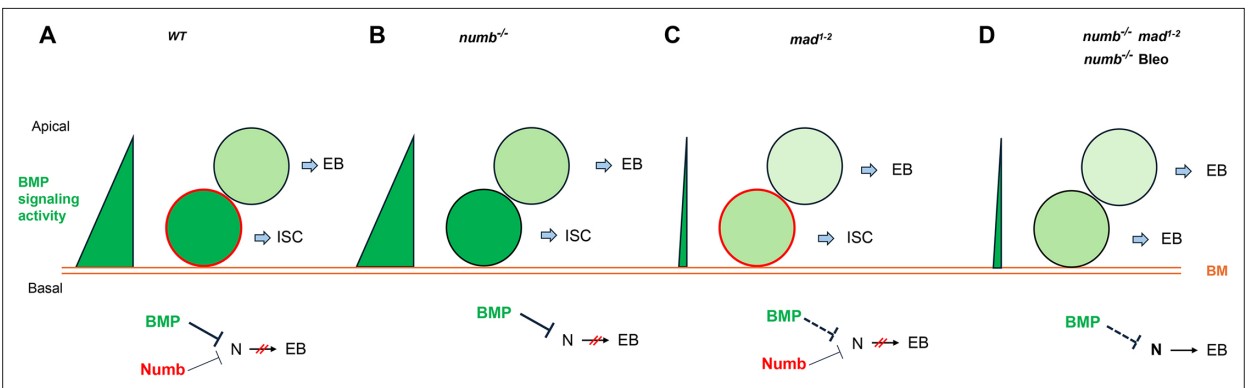

**Figure 7.** Model for Numb and BMP signaling in intestinal stem cell (ISC)/enteroblast (EB) fate decision. (**A**) During asymmetric ISC division, the basal ISC daughter transduces higher level of BMP signaling and inherits higher level of Numb activity than the apical one. Inhibition of N by BMP signaling and Numb promotes ISC fate. (**B**) In *numb* mutant background, differential BMP signaling between the basal and apical ISC daughters is sufficient to generate differential N pathway activities to drive asymmetric fate decision. (**C**) In *mad* mutant background, the shallow BMP activity gradient acts in conjunction with the asymmetric Numb activity to generate differential N pathway activities between the basal and apical ISC daughters to drive asymmetric fate decision. (**D**) In *numb mad* double mutant background or in guts containing *numb* mutant clones and injured by bleomycin (Bleo) feeding, the shallow BMP activity gradient is often insufficient to generate asymmetric N pathway activation, leading to precocious ISC-to-EB differentiation. BM: basement membrane; Bleo: bleomycin; thin and dashed lines indicate weak inhibition.

# Materials and methods

## Key resources table

| Reagent type (species) or resource | Designation | Source or reference | Identifiers | Additional information |
|---|---|---|---|---|
| Genetic reagent (*D. melanogaster*) | esg-Gal4 | *Jiang et al., 2009* | FLYB: FBti0013268 | FlyBase symbol: P{GawB}NP5130 |
| Genetic reagent (*D. melanogaster*) | Dl-lacZ | *Zeng et al., 2010* | FLYB: FBti0004778 | FlyBase symbol: P{PZ}Delta[05151] |
| Genetic reagent (*D. melanogaster*) | Su(H)-lacZ | *Zeng et al., 2010* | FLYB: FBtp0014034 | FlyBase symbol: P{Ddc.E(spl)m8-HLH-lacZ.Gbe} |
| Genetic reagent (*D. melanogaster*) | E(spl)mβCD2 | Bloomington *Drosophila* Stock Center | BDSC 83353 FLYB: FBst0083353 RRID:BDSC_83353 | FlyBase symbol: w[*]; l(2)*[*]/CyO, P{ry[+t7.2]=en1} wg[en11]; P{w[+mC]=E(spl)mbeta-HLH-CD2.dC}T6 |
| Genetic reagent (*D. melanogaster*) | Tub-Gal80[ts] | *Jiang et al., 2009* | FLYB: FBti0027796 | FlyBase symbol: P{tubP-GAL80ts} |
| Genetic reagent (*D. melanogaster*) | UAS-Put RNAi | Vienna *Drosophila* Resource Center | VDRC 107071 FLYB: FBst0478894 RRID:Flybase_FBst0473060 | FlyBase symbol: P{KK102676}VIE-260B |
| Genetic reagent (*D. melanogaster*) | UAS-Dl RNAi | Bloomington *Drosophila* Stock Center | BDSC 28032 FLYB: FBst0028032 RRID:BDSC_28032 | FlyBase symbol: y (1) v(1); P{TRiP.JF02867}attP2 |
| Genetic reagent (*D. melanogaster*) | UAS-Mad RNAi | Vienna *Drosophila* Resource Center | VDRC 12635 FLYB: FBst0450590 RRID:Flybase_FBst0450590 | FlyBase symbol: w1118; P{GD4121}v12635 |
| Genetic reagent (*D. melanogaster*) | UAS-Numb RANi | Bloomington *Drosophila* Stock Center | BDSC 35045 FLYB: FBst0035045 RRID:BDSC_35045 | FlyBase symbol: y(1) sc[*] v(1) sev(21); P{y[+t7.7] v[+t1.8]=TRiP.HMS01459}attP2 |
| Genetic reagent (*D. melanogaster*) | UAS-mCherry RANi | Bloomington *Drosophila* Stock Center | BDSC 35785 FLYB: FBst0035785 RRID:BDSC_35785 | FlyBase symbol: y(1) sc[*] v(1) sev(21); P{y[+t7.7] v[+t1.8]=VALIUM20-mCherry.RNAi}attP2 |
| Genetic reagent (*D. melanogaster*) | numb[4] | *Skeath and Doe, 1998* | FLYB: FBal0090215 | FlyBase symbol: numb(4) |
| Genetic reagent (*D. melanogaster*) | numb[15] | *Sallé et al., 2017* | FLYB: FBal0146969 | FlyBase symbol: numb(15) |
| Genetic reagent (*D. melanogaster*) | mad[1-2] | Bloomington *Drosophila* Stock Center | BDSC 7323 FLYB: FBst0007323 RRID:BDSC_7323 | FlyBase symbol: w*; Mad(; 1-2) P{neoFRT}40A/CyO |
| Genetic reagent (*D. melanogaster*) | FRT 40A | Bloomington *Drosophila* Stock Center | BDSC 8212 FLYB: FBst0008212 RRID:BDSC_8212 | FlyBase symbol: w[1118]; P{ry[+t7.2]=neoFRT}40A/CyO; P{ry[+t7.2]=ey-FLP.N}6, ry[506] |
| Genetic reagent (*D. melanogaster*) | yw, hs-FLP | Bloomington *Drosophila* Stock Center | BDSC 1929 FLYB: FBst0001929 RRID:BDSC_1929 | FlyBase symbol: P{ry[+t7.2]=hsFLP}12, y(1) w[*]; sna[Sco]/CyO |
| Genetic reagent (*D. melanogaster*) | FRT 40A, tub-GAL80 | Bloomington *Drosophila* Stock Center | BDSC 5192 FLYB: FBst0005192 RRID:BDSC_5192 | FlyBase symbol: y(1) w[*]; P{w[+mC]=tubP-GAL80} LL10 P{ry[+t7.2]=neoFRT}40A/CyO |
| Genetic reagent (*D. melanogaster*) | UAS-GFP | Bloomington *Drosophila* Stock Center | BDSC 5130 FLYB: FBst0005130 RRID:BDSC_5130 | FlyBase symbol: y(1) w[*]; betaTub60D[Pin-Yt]/CyO; P{w[+mC]=UAS-mCD8::GFP.L}LL6 |
| Genetic reagent (*D. melanogaster*) | FRT 19A | Bloomington *Drosophila* Stock Center | BDSC 1709 FLYB: FBst0001709 RRID:BDSC_1709 | FlyBase symbol: P{ry[+t7.2]=neoFRT}19A; ry[506] |
| Genetic reagent (*D. melanogaster*) | Ubi-GFPnls | *Chen and Schüpbach, 2006* | FLYB: FBti0015575 | FlyBase symbol: P{Ubi-GFP(S65T)nls}X |

*Continued on next page*

*Continued*

| Reagent type (species) or resource | Designation | Source or reference | Identifiers | Additional information |
|---|---|---|---|---|
| Genetic reagent (*D. melanogaster*) | FRT 19A, ubi-mRFPnls | Bloomington *Drosophila* Stock Center | BDSC 31418 FLYB: FBst0031418 RRID:BDSC_31418 | FlyBase symbol: P{w[+mC]=Ubi-mRFP.nls}1, w[*], P{ry[+t7.2]=hsFLP}12P{ry[+t7.2]=neoFRT}19A |
| Antibody | anti-GFP (chicken polyclonal) | Abcam | Cat#: ab13970; RRID:AB_300798 | IF (1:1000) |
| Antibody | anti-β-Galactosidase (rabbit polyclonal) | MP Biomedicals | Cat#: 08559761 RRID:AB_3675281 | IF (1:1000) |
| Antibody | anti-Rat CD2 (mouse monoclonal) | Bio-Rad | Cat#: MCA154GA RRID:AB_566608 | IF (1:2000) (Formerly AbD Serotec) |
| Antibody | anti-Dl extracellular domain (mouse monoclonal) | DSHB | Cat#: c594.9b RRID:AB_528194 | IF (1:20) |
| Antibody | anti-Prospero (mouse monoclonal) | DSHB | Cat#: Prospero RRID:AB_528440 | IF (1:20) |
| Antibody | anti-Pdm1 (rabbit polyclonal) | Dr. Xiaohang Yang | | IF (1:1000) |
| Antibody | anti-Chicken Alexa Fluor 488 (goat polyclonal secondary) | Thermo Fisher Scientific | Cat#: A-11039 RRID:AB_2534096 | IF (1:1000) |
| Antibody | anti-Mouse Alexa Flour 546 (goat polyclonal secondary) | Thermo Fisher Scientific | Cat#: A-11030 RRID:AB_2737024 | IF (1:1000) |
| Antibody | anti-Rabbit Alexa Flour 546 (goat polyclonal secondary) | Thermo Fisher Scientific | Cat#: A-11035 RRID:AB_2534093 | IF (1:1000) |
| Antibody | anti-Mouse Cy5 (goat polyclonal) | Jackson ImmunoResearch Labs | Cat#: 115-175-166 RRID:AB_2338714 | IF (1:500) |
| Antibody | anti-Rabbit Cy5 (goat polyclonal) | Jackson ImmunoResearch Labs | Cat#: 111-175-144 RRID:AB_2338013 | IF (1:500) |
| Chemical compound, drug | DAPI (4',6-diamidino-2-phenylindole, dihydrochloride) | Invitrogen | Cat#: D1306 | IF (1:2000) |
| Chemical compound, drug | DSS (dextran sulfate sodium) | Sigma-Aldrich | Cat#: 42867 | 5% solution |
| Chemical compound, drug | Bleomycin sulfate from *Streptomyces verticillus* | Sigma-Aldrich | Cat#: B8416 | 25 µg/ml |
| Chemical compound, drug | Sucrose | Fisher Bioreagent | Cat#: BP220-212 | 5% solution |

### *Drosophila* genetics and transgenes

Flies were maintained on cornmeal at 25°C. Transgenic lines and mutants include: *UAS-Put-RNAi* (VDRC #107071); *UAS-Dl-RNAi* (BL#28032); *UAS-Mad-RNAi* (VDRC #12635); *UAS-Numb-RNAi* (BL #35045); *UAS-mCherry-RNAi* (BL #35785); *tub-Gal80^ts^*, *esg-Gal4, Su(H) Gbe-lacZ (Su(H)-lacZ)*; *E(spl) mβ-CD2* (BL#83353); *numb^4^* is a strong allele, *numb^15^* is a null allele, and *mad^1-2^* is a hypomorphic allele (Flybase). *yw hs-FLP UAS-GFP*; *tub-Gal80 FRT40A* was used for MARCM clonal analysis. *yw hs-FLP FRT19A ubi-GFPnls* and *yw FRT19A ubi-mRFPnls* were used for twin-spot clone analysis. For experiments involving *tubGal80^ts^*, crosses were set up and cultured at 18°C to restrict Gal4 activity. Two- to three-day-old progenies were shifted to 29°C for the indicated periods of time to inactivate *Gal80^ts^*, allowing Gal4 to activate *UAS* transgenes in all experiments, only the female posterior midguts were analyzed.

## Genotypes for *Drosophila* used in each figure

### *Figure 1*

(B–B") *Su(H)-lacZ/+; esg-Gal4 tub-Gal80ts/+*
(C–C") *Su(H)-lacZ/+; esg-Gal4 tub-Gal80ts/UAS-Put-RNAi*
(D–D") *Su(H)-lacZ/+; esg-Gal4 tub-Gal80ts/+; UAS-Dl-RNAi/+*
(E–E") *Su(H)-lacZ/+; esg-Gal4 tub-Gal80ts/UAS-Put-RNAi; UAS-Dl-RNAi/+.*

### *Figure 2*

(A–A") *yw/+; esg-Gal4 tub-Gal80ts/+; UAS-mCherry-RNAi, E(spl)mβ-CD2/Dl-lacZ*
(B–B") *yw/+; esg-Gal4 tub-Gal80ts/UAS-Mad-RNAi; E(spl)mβ-CD2/Dl-lacZ*
(C–C") *yw/+; esg-Gal4 tub-Gal80ts/+; UAS-Numb-RNAi, E(spl)mβ-CD2/Dl-lacZ*
(D–D") *yw/+; esg-Gal4 tub-Gal80ts/UAS-Mad-RNAi; UAS-Numb-RNAi, E(spl)mβ-CD2/Dl-lacZ.*

### *Figure 3*

(A–A') *yw,hs-FLP/+; FRT40A tub-GAL80/FRT40A; UAS-GFP/+*
(B–B') *yw,hs-FLP/+; FRT40A tub-GAL80/FRT40A mad1-2; UAS-GFP/+*
(C–C') *yw,hs-FLP/+; FRT40A tub-GAL80/FRT40A numb4; UAS-GFP/+*
(D–D') *yw,hs-FLP/+; FRT40 tub-GAL80/FRT40A mad1-2,numb4; UAS-GFP/+*
(E–E', I-I') *yw,hs-FLP/+; FRT40A tub-GAL80/ FRT40A; UAS-GFP/E(spl)mβ-CD2*
(F–F') *yw,hs-FLP/+; FRT40A tub-GAL80/FRT40A mad1-2; UAS-GFP/E(spl)mβ-CD2*
(G–G') *yw,hs-FLP/+; FRT40A tub-GAL80/ FRT40A numb; UAS-GFP/E(spl)mβ-CD2*
(H–H', J–J') *yw,hs-FLP/+; FRT40A tub-GAL80/ FRT40A mad1-2 ,numb4; UAS-GFP/E(spl)mβ-CD2.*

### *Figure 4*

(D–F) *yw hs-FLP, FRT19A ubi-GFPnls/FRT19A ubi-mRFPnls; esg-Gal4 tub-Gal80ts/+*
(G–I) *yw hs-FLP, FRT19A ubi-GFPnls/FRT19A ubi-mRFPnls; esg-Gal4 tub-Gal80ts/UAS-Mad-RNAi*
(J–L) *yw hs-FLP, FRT19A ubi-GFPnls/FRT19A ubi-mRFPnls; esg-Gal4 tub-Gal80ts/+; UAS-Numb-RNAi/+*
(M–O) *yw hs-FLP, FRT19A ubi-GFPnls/FRT19A ubi-mRFPnls; esg-Gal4 tub-Gal80ts/UAS-Mad-RNAi; UAS-Numb-RNAi/+.*

### *Figure 5*, *Figure 5—figure supplement 1*

(A–A') *yw hs-FLP/+; FRT40A tub-GAL80/FRT40A; UAS-GFP/+*
(B–B') *yw hs-FLP/+; FRT40A tub-GAL80/FRT40A numb4 382 ; UAS-GFP/+*
(C–C') *yw hs-FLP/+; FRT40A tub-GAL80/FRT40A numb15 383 ; UAS-GFP/+.*

### *Figure 6*

(A-C') *yw hs-FLP/+; FRT40A tub-GAL80/FRT40A; UAS-GFP/+*
(D-F') *yw hs-FLP/+; FRT40A tub-GAL80/FRT40A numb4; UAS-GFP/+*
(F-F') *yw,hs-FLP/+; FRT40A tub-GAL80/FRT40A mad1-2; UAS-GFP/ E(spl)mβ-CD2.* (G-G') *yw,hs-FLP/+; FRT40A tub-GAL80/FRT40A numb; UAS-GFP/E(spl)mβ-CD2.*

## MARCM clone analysis

For MARCM clone induction, crosses were set up and cultured at 18°C to avoid spontaneous clones. 2-to-3-day-old females were subjected to heat shock at 37°C for 1 hr and then kept at 18°C for another 14 days before dissection. Flies were transferred to new vials with fresh food every 2 days. The sizes of the clones were quantified from at least 10 midguts for each genotype.

## Twin-spot clone analysis

For twin-spot clone generation, 2- to 3-day-old flies were kept at 29°C for 14 days and heat-shocked at 37°C for 1 hr and then raised at 29°C for another 4 days before dissection. Flies were transferred to new vials with fresh food every 2 days.

## Feeding experiments

Flies were cultured in an empty vial containing a piece of 2.5 × 3.75 cm chromatography paper (Fisher) wet with 5% sucrose (MP Biomedicals) solution as feeding medium (mock treatment) or with 25 μg/ml bleomycin (Sigma-Aldrich) or 5% DSS (40 kDa; MP Biomedicals) for 1 day at 30°C. After treatment, flies were recovered on normal food at 18°C for another 4 days before dissection.

## Immunostaining

Female flies were used for gut immunostaining in all experiments. The entire gastrointestinal tract was taken and fixed in 1× PBS plus 8% EM grade formaldehyde (Polysciences) for 2 hr. Samples were washed and incubated with primary and secondary antibodies in a solution containing 1× PBS, 0.5% goat serum (Thermo Fisher), and 0.1% Triton X-100 (Bio-rad). The following primary antibodies were used: mouse anti-Delta (DSHB), 1:10; rabbit anti-LacZ (MP Biomedicals), 1:1000; mouse anti-CD2 (Thermo Fisher), 1:1000; chicken anti-GFP (Abcam), 1:1000; mouse anti-Pros (DSHB), 1:10; rabbit anti-Pdm1 (from Dr. Xiaohang Yang), 1:1000; Alexa Fluor-conjugated secondary antibodies were used at 1:1000 (Invitrogen). DAPI (4',6-diamidino-2-phenylindole) is a nuclear dye (Thermo Fisher). Guts were mounted in 70% glycerol and imaged with a Zeiss confocal microscope (Zeiss LSM 710 inverted confocal) using ×40 oil objectives (imaging medium: Zeiss Immersol 518F). The acquisition and processing software was Zeiss LSM Image Browser, and image processing was done in Adobe Photoshop.

## Quantification and statistical analysis

For *Figures 1 and 2*, cell number of the indicated cell types was counted per ROI (region of interest) on images taken using LEICA DFC345 FX camera on a LEICA DMI 400 B microscope, equipped with a ×40 objective lens. For each genotype, 8–12 guts were analyzed. In each gut, three ROIs were randomly selected in R4 of midguts for quantification. One-way ANOVA was performed for statistical comparisons. For *Figures 3–6*, all GFP$^+$ clone cells (≥2) in midguts were counted individually. For each genotype, at least 10 guts were calculated. $x^2$ test was performed for statistical comparisons. All statistical significances were calculated in Prism 10 (GraphPad Software, Inc). *$p < 0.05$, **$p < 0.01$, ***$p < 0.001$, ****$p < 0.0001$; n.s., not significant.

## Acknowledgements

We thank Bing Wang for technical assistance, Vienna *Drosophila* Resource Center and Bloomington *Drosophila* Stock Center for fly stocks, and Developmental Studies Hybridoma Bank for antibodies.

## Additional information

### Funding

| Funder | Grant reference number | Author |
|---|---|---|
| National Institute of General Medical Sciences | GM118063 | Jin Jiang |
| Welch Foundation | I-1603 | Jin Jiang |

The funders had no role in study design, data collection, and interpretation, or the decision to submit the work for publication.

### Author contributions

Mengjie Li, Data curation, Formal analysis, Validation, Investigation, Visualization, Methodology, Writing – original draft; Aiguo Tian, Data curation, Formal analysis, Visualization, Methodology; Jin

Jiang, Conceptualization, Supervision, Funding acquisition, Writing – original draft, Project administration, Writing – review and editing

**Author ORCIDs**
Mengjie Li (iD) http://orcid.org/0000-0001-7924-5331
Aiguo Tian (iD) https://orcid.org/0000-0001-5011-6034
Jin Jiang (iD) https://orcid.org/0000-0002-5951-372X

Reviewer #1 (Public review): https://doi.org/10.7554/eLife.104723.3.sa1
Reviewer #2 (Public review): https://doi.org/10.7554/eLife.104723.3.sa2
Reviewer #3 (Public review): https://doi.org/10.7554/eLife.104723.3.sa3
Author response https://doi.org/10.7554/eLife.104723.3.sa4

---

## Additional files

### Supplementary files
MDAR checklist

### Data availability
All data generated or analyses during this study are included in the manuscript and supporting files. Source data have been provided for Figures 1–6.

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
