## [Editor Report · eLife Assessment]

The authors examine the role of Numb, a Notch inhibitor, in intestinal stem cell self-renewal in *Drosophila* during homeostasis and regeneration. The significance is **important** as the authors demonstrate the ISC maintenance is reduced when both BMP signaling and Numb expression is reduced. The strength of evidence is **convincing** as large sample sizes and statistical analyses are provided.

---

## [Referee Report · Reviewer #1 (Public review)]

Summary:

By way of background, the Jiang lab has previously shown that loss of the type II BMP receptor Punt (Put) from intestinal progenitors (ISCs and EBs) caused them to differentiate into EBs, with a concomitant loss of ISCs (Tian and Jiang, eLife 2014). The mechanism by which this occurs was activation of Notch in Put-deficient progenitors. How Notch was upregulated in Put-deficient ISCs was not established in this prior work. In the current study, the authors test whether a very low level of Dl was responsible. But co-depletion of Dl and Put led to a similar phenotype as depletion of Put alone. This result suggested that Dl was not the mechanism. They next investigate genetic interactions between BMP signaling and Numb, an inhibitor of Notch signaling. Prior work from Bardin, Schweisguth and other labs has shown that Numb is not required for ISC self-renewal. But the authors wanted to know whether loss of both the BMP signal transducer Mad and Numb would cause ISC loss. This result was observed for RNAi depletion from progenitors and for mad, numb double mutant clones. Of note, ISC loss was observed in 40% of mad, numb double mutant clones, whereas 60% of these clones had an ISC. They then employed a two-color tracing system called RGT to look at the outcome of ISC divisions (asymmetric (ISC/EB) or symmetric (ISC/ISC or EB/EB)). Control clones had 69%, 15% and 16%, respectively, whereas mad, numb double mutant clones had much lower ISC/ISC (11%) and much higher EB/EB (37%). They conclude that loss of Numb in moderate BMP loss of function mutants increased symmetric differentiation which lead caused ISC loss. They also reported that numb15 and numb4 clones had a moderate but significant increase in ISC-lacking clones compared to control clones, supporting the model that Numb plays a role in ISC maintenance. Finally, they investigated the relevance of these observation during regeneration. After bleomycin treatment, there was a significant increase in ISC-lacking clones and a significant decrease in clone size in numb4 and numb15 clones compared to control clones. Because bleomycin treatment has been shown to cause variation in BMP ligand production, the authors interpret the numb clone under bleomycin results as demonstrating an essential role of Numb in ISC maintenance during regeneration.

Strengths

i. Data are quantified with statistical analysis

ii. Experiments have appropriate controls and large numbers of samples

iii. Results demonstrate an important role of Numb in maintaining ISC number during regeneration and a genetic interaction between Mad and Numb during homeostasis.

Weaknesses

None noted in the revised manuscript.

---

## [Referee Report · Reviewer #2 (Public review)]

Summary:

This work assesses the genetic interaction between the Bmp signaling pathway and the factor Numb, which can inhibit Notch signalling. It follows up on the previous studies of the group (Tian, eLife, 2014; Tian, PNAS, 2014) regarding BMP signaling in controlling stem cell fate decision as well as on the work of another group (Sallé, EMBO, 2017) that investigated the function of Numb on enteroendocrine fate in the midgut. This is an important study providing evidence of a Numb-mediated back up mechanism for stem cell maintenance.

Strengths:

(1) Experiments are consistent with these previous publications while also extending our understanding of how Numb functions in the ISC.

(2) Provides an interesting model of a "back up" protection mechanism for ISC maintenance.

Weaknesses:

(1) Aspects of the experiments could be better controlled or annotated:

(a) As they "randomly chose" the regions analyzed, it would be better to have all from a defined region (R4 or R2, for example) or to at least note the region as there are important regional differences for some aspects of midgut biology.

(b) It is not clear to me why MARCM clones were induced and then flies grown at 18{degree sign}C? It would help to explain why they used this unconventional protocol.

(2) There are technical limitations with trying to conclude from double-knockdown experiments in the ISC lineage, such as those in Figure 1 where Dl and put are both being knocked down: depending on how fast both proteins are depleted, it may be that only one of them (put, for example) is inactivated and affects the fate decision prior to the other one (Dl) being depleted. Therefore, it is difficult to definitively conclude that the decision is independent of Dl ligand.

(3) Additional quantification of many phenotypes would be desired.

(a) It would be useful to see esg-GFP cells/total cells and not just field as the density might change (2E for example).

(b) Similarly, for 2F and 2G, it would be nice to see the % of ISC/ total cell and EB/total cell and not only per esgGFP+ cell.

(c) Fig1: There is no quantification - specifically it would be interesting to know how many esg+ are su(H)lacZ positive in Put- Dl- condition compared to WT or Put- alone. What is the n?

(d) Fig2: Pros + cells are not seen in the image? Are they all DllacZ+?

(e) Fig3: it would be nice to have the size clone quantification instead of the distribution between groups of 2 cell 3 cells 4 cell clones.

(f) How many times were experiments performed?

(4) The authors do not comment on the reduction of clone size in DSS treatment in Figure 6K. How do they interpret this? Does it conflict with their model of Bleo vs DSS?

(5) There is probably a mistake on sentence line 314 -316 "Indeed, previous studies indicate that endogenous Numb was not undetectable by Numb antibodies that could detect Numb expression in the nervous system".

Comments on revisions:

The authors have by and large addressed my main points.

---

## [Referee Report · Reviewer #3 (Public review)]

Summary:

The authors provide an in-depth analysis of the function of Numb in adult *Drosophila* midgut. Based on RNAi combinations and double mutant clonal analyses, they propose that Numb has a function in inhibiting Notch pathway to maintain intestinal stem cells, and is a backup mechanism with BMP pathway in maintaining midgut stem cell mediated homeostasis.

Strengths:

Overall, this is a carefully constructed series of experiments, and the results and statistical analyses provides believable evidence that Numb has a role, albeit weak compared to other pathways, in sustaining ISC and in promoting regeneration especially after damage by bleomycin, which may damage enterocytes and therefore disrupt BMP pathway more. The results overall support their claim.

The data are highly coherent, and support a genetic function of Numb, in collaborating with BMP signaling, to maintain the number and proliferative function of ISCs in adult midguts. The authors used appropriate and sophisticated genetic tools of double RNAi, mutant clonal analysis and dual marker stem cell tracing approaches to ensure the results are reproducible and consistent. The statistical analyses provide confidence that the phenotypic changes are reliable albeit weaker than many other mutants previously studied.

Weaknesses:

In the absence of Numb itself, the midgut has a weak reduction of ISC number (Fig. 3 and 5), as well as weak albeit not statistically significant reduction of ISC clone size/proliferation. I think the authors published similar experiments with BMP pathway mutants. The mad1-2 allele used here as stated below may not be very representative of other BMP pathway mutants. Therefore, it could be beneficial to compare the number of ISC number and clone sizes between other BMP experiments to provide the readers a clearer picture how these two pathways individually contribute (stronger/weaker effects) to the ISC number and gut homeostasis.

The main weakness of this manuscript is the analysis of the BMP pathway components, especially the mad1-2 allele. The mad RNAi and mad1-2 alleles (P insertion) are supposed to be weak alleles and that might be suitable for genetic enhancement assays here together with numb RNAi. However, the mad1-2 allele, and sometime the mad RNAi, showed weakly increased ISC clone size. This is kind of counter-intuitive that they should have a similar ISC loss and ISC clone size reduction.

A much stronger phenotype was observed when numb mutants were subject to treatment of tissue damaging agents Bleomycin, which causes damage in different ways than DSS. Bleomycin as previously shown to be causing mainly enterocyte damage, and therefore disrupt BMP signaling from ECs more likely. Therefore, this treatment together with loss of numb led to highly significant reduction of ISC in clones and reduction of clone size/proliferation. One improvement is that it is not clear whether the authors discussed the nature of the two numb mutant alleles used in this study and the comparison to the strength of the RNAi allele. Because the phenotypes are weak, and more variable, the use of specific reagents is important.

Furthermore, the use of possible activating alleles of either or both pathways to test genetic enhancement or synergistic activation will provide strong support for the claims.

For the revision, the authors have provided detailed responses, comments, and a revised manuscript that together satisfactorily answer all my questions. The manuscript read well and the flow of information is quite clear. I do not have further concerns and support the manuscript moving forward.

---

## [Author Response]

The following is the authors’ response to the original reviews

**Reviewer #1 (Public review):**
Summary:By way of background, the Jiang lab has previously shown that loss of the type II BMP receptor Punt (Put) from intestinal progenitors (ISCs and EBs) caused them to differentiate into EBs, with a concomitant loss of ISCs (Tian and Jiang, eLife 2014). The mechanism by which this occurs was activation of Notch in Put-deficient progenitors. How Notch was upregulated in Put-deficient ISCs was not established in this prior work. In the current study, the authors test whether a very low level of Dl was responsible. But co-depletion of Dl and Put led to a similar phenotype as depletion of Put alone. This result suggested that Dl was not the mechanism. They next investigate genetic interactions between BMP signaling and Numb, an inhibitor of Notch signaling. Prior work from Bardin, Schweisguth and other labs has shown that Numb is not required for ISC self-renewal. However the authors wanted to know whether loss of both the BMP signal transducer Mad and Numb would cause ISC loss. This result was observed for RNAi depletion from progenitors and for mad, numb double mutant clones. Of note, ISC loss was observed in 40% of mad, numb double mutant clones, whereas 60% of these clones had an ISC. They then employed a two-color tracing system called RGT to look at the outcome of ISC divisions (asymmetric (ISC/EB) or symmetric (ISC/ISC or EB/EB)). Control clones had 69%, 15% and 16%, respectively, whereas mad, numb double mutant clones had much lower ISC/ISC (11%) and much higher EB/EB (37%). They conclude that loss of Numb in moderate BMP loss of function mutants increased symmetric differentiation which lead caused ISC loss. They also reported that numb^15^ and numb^4^ clones had a moderate but significant increase in ISC-lacking clones compared to control clones, supporting the model that Numb plays a role in ISC maintenance. Finally, they investigated the relevance of these observation during regeneration. After bleomycin treatment, there was a significant increase in ISC-lacking clones and a significant decrease in clone size in numb^4^ and numb^15^ clones compared to control clones. Because bleomycin treatment has been shown to cause variation in BMP ligand production, the authors interpret the numb clone under bleomycin results as demonstrating an essential role of Numb in ISC maintenance during regeneration.Strengths:(i) Most data is quantified with statistical analysis(ii) Experiments have appropriate controls and large numbers of samples(iii) Results demonstrate an important role of Numb in maintaining ISC number during regeneration and a genetic interaction between Mad and Numb during homeostasis.Weaknesses:(i) No quantification for Fig. 1

Quantification of Fig.1 has been added.

(ii) The premise is a bit unclear. Under homeostasis, strong loss of BMP (Put) leads to loss of ISCs, presumably regardless of Numb level (which was not tested). But moderate loss of BMP (Mad) does not show ISC loss unless Numb is also reduced. I am confused as to why numb does not play a role in Put mutants. Did the authors test whether concomitant loss of Put and Numb leads to even more ISC loss than Put-mutation alone.

We have tested the genetic interaction between put and numb using Put RNAi and Numb RNAi driven by esg^ts^. According to the results in this study and our previously published data, *put* mutant clone or *esgts* > *Put-RNAi* induced a rapid loss of ISC (whin 8 days). We did not observe further enhancement of stem cell loss phenotype in Put and Numb double RNAi guts.

(iii) I think that the use of the word "essential" is a bit strong here. Numb plays an important role but in either during homeostasis or regeneration, most numb clones or mad, numb double mutant clones still have ISCs. Therefore, I think that the authors should temper their language about the role of Numb in ISC maintenance.

We have revised the language and changed “essential” to important”.

**Reviewer #2 (Public review):**
Summary:This work assesses the genetic interaction between the Bmp signaling pathway and the factor Numb, which can inhibit Notch signalling. It follows up on the previous studies of the group (Tian, Elife, 2014; Tian, PNAS, 2014) regarding BMP signaling in controlling stem cell fate decision as well as on the work of another group (Sallé, EMBO, 2017) that investigated the function of Numb on enteroendocrine fate in the midgut. This is an important study providing evidence of a Numb-mediated back up mechanism for stem cell maintenance.Strengths:(1) Experiments are consistent with these previous publications while also extending our understanding of how Numb functions in the ISC.(2) Provides an interesting model of a "back up" protection mechanism for ISC maintenance.Weaknesses:(1) Aspects of the experiments could be better controlled or annotated:(a) As they "randomly chose" the regions analyzed, it would be better to have all from a defined region (R4 or R2, for example) or to at least note the region as there are important regional differences for some aspects of midgut biology.

Thank you for the suggestion. In fact, we conducted all the analyses in region 4, we have added statement to clarify this in the revised manuscript.

(b) It is not clear to me why MARCM clones were induced and then flies grown at 18{degree sign}C? It would help to explain why they used this unconventional protocol.

We kept the flies at 18°C to avoid spontaneous clone.

(2) There are technical limitations with trying to conclude from double-knockdown experiments in the ISC lineage, such as those in Figure 1 where Dl and put are both being knocked down: depending on how fast both proteins are depleted, it may be that only one of them (put, for example) is inactivated and affects the fate decision prior to the other one (Dl) being depleted. Therefore, it is difficult to definitively conclude that the decision is independent of Dl ligand.

In our hand, Dl-RNAi is very effective and exhibited loss of N pathway activity (as determined by the N pathway reporter Su(H)-lacZ) after RNAi for 8 days (Fig. 1D). Therefore, the ectopic Su(H)-lacZ expression in Punt Dl double RNAi (fig. 1E) is unlikely due to residual Dl expression. Nevertheless, we have changed the statement “BMP signaling blocks ligand-independent N activity” to” Loss of BMP signaling results in ectopic N pathway activity even when Dl is depleted”

(3) Additional quantification of many phenotypes would be desired.(a) It would be useful to see esg-GFP cells/total cells and not just field as the density might change (2E for example).

We focused on R4 region for quantification where the cell density did not exhibit apparent change in different experimental groups. In addition, we have examined many guts for quantification. It is very unlikely that the difference in the esg-GFP+ cell number is caused by change in cell density.

(b) Similarly, for 2F and 2G, it would be nice to see the % of ISC/ total cell and EB/total cell and not only per esgGFP+ cell.

Unfortunately, we didn’t have the suggested quantification. However, we believe that quantification of the percentage of ISC or EB among all progenitor cells, as we did here, provides a meaningful measurement of the self-renewal status of each experimental group.

(c) Fig1: There is no quantification - specifically it would be interesting to know how many esg+ are su(H)lacZ positive in Put- Dl- condition compared to WT or Put- alone. What is the n?

Quantification of Fig.1 has been added.

(d) Fig2: Pros + cells are not seen in the image? Are they all DllacZ+?

Anti-Pros and anti-E(spl)mβ-CD2 were stained in the same channel (magenta). Pros+ exhibited “dot-like” nuclear staining while CD2 staining outlined the cell membrane of EBs. We have clarified this in the revised figure legend.

(e) Fig3: it would be nice to have the size clone quantification instead of the distribution between groups of 2 cell 3 cells 4 cell clones.

Because of the heterogeneity of clone size for each genotype, we chose to group clones based on their sizes (2, 3-6, 6-8, >8 cells) and quantified the distribution of individual groups for each genotype, which clearly showed an overall reduction in clone size for mad numb double mutant clones. We and others have used the same clone size analysis in previous studies (e.g., Tian and Jiang, eLife 2014).

(f) How many times were experiments performed?

All experiments were performed at least 3 times.

(4) The authors do not comment on the reduction of clone size in DSS treatment in Figure 6K. How do they interpret this? Does it conflict with their model of Bleo vs DSS?

Guts containing numb^4^ clones treated with DSS exhibited a slight reduction of clone size, evident by a higher percentage of 2-cell clones and lower percentage of > 8 cell clones. This reduction is less significant in guts containing numb^15^ clones. However, the percentage of Dl^+^-containing clones is similar between DSS and mock-treated guts. It is possible that ISC proliferation is lightly reduced due to numb^4^ mutation or the genetic background of this stock.

(5) There is probably a mistake on sentence line 314 -316 "Indeed, previous studies indicate that endogenous Numb was not undetectable by Numb antibodies that could detect Numb expression in the nervous system".

We have modified the sentence.

**Reviewer #3 (Public review):**
Summary:The authors provide an in-depth analysis of the function of Numb in adult *Drosophila* midgut. Based on RNAi combinations and double mutant clonal analyses, they propose that Numb has a function in inhibiting Notch pathway to maintain intestinal stem cells, and is a backup mechanism with BMP pathway in maintaining midgut stem cell mediated homeostasis.Strengths:Overall, this is a carefully constructed series of experiments, and the results and statistical analyses provides believable evidence that Numb has a role, albeit weak compared to other pathways, in sustaining ISC and in promoting regeneration especially after damage by bleomycin, which may damage enterocytes and therefore disrupt BMP pathway more. The results overall support their claim.The data are highly coherent, and support a genetic function of Numb, in collaborating with BMP signaling, to maintain the number and proliferative function of ISCs in adult midguts. The authors used appropriate and sophisticated genetic tools of double RNAi, mutant clonal analysis and dual marker stem cell tracing approaches to ensure the results are reproducible and consistent. The statistical analyses provide confidence that the phenotypic changes are reliable albeit weaker than many other mutants previously studied.Weaknesses:In the absence of Numb itself, the midgut has a weak reduction of ISC number (Fig. 3 and 5), as well as weak albeit not statistically significant reduction of ISC clone size/proliferation. I think the authors published similar experiments with BMP pathway mutants. The mad^1-2^ allele used here as stated below may not be very representative of other BMP pathway mutants. Therefore, it could be beneficial to compare the number of ISC number and clone sizes between other BMP experiments to provide the readers with a clearer picture of how these two pathways individually contribute (stronger/weaker effects) to the ISC number and gut homeostasis.

Thanks for the comment. We have tested other components of BMP pathway in our previously study (Tian et al., 2014). More complete loss of BMP signaling (for example, Put clones, Put RNAi, Tkv/Sax double mutant clones or double RNAi) resulted in ISC loss regardless the status of numb, suggesting a more predominant role of BMP signaling in ISC self-renewal compared with Numb. We speculate that the weak stem cell loss phenotype associated with numb mutant clones in otherwise wild type background could be due to fluctuation of BMP signaling in homeostatic guts.

The main weakness of this manuscript is the analysis of the BMP pathway components, especially the mad^1-2^ allele. The mad RNAi and mad^1-2^ alleles (P insertion) are supposed to be weak alleles and that might be suitable for genetic enhancement assays here together with numb RNAi. However, the mad^1-2^ allele, and sometimes the mad RNAi, showed weakly increased ISC clone size. This is kind of counter-intuitive that they should have a similar ISC loss and ISC clone size reduction.

We used mad^1-2^ and mad RNAi here to test the genetic interaction with numb because our previous studies showed that partial loss of BMP signaling under these conditions did not cause stem cell loss, therefore, may provide a sensitized background to determine the role of Numb in ISC self-renewal. The increased proliferation of ISC/ clone size associated with mad^1-2^ and mad RNAi is due to the fact that reduction of BMP signaling in either EC or EB non-autonomously induces stem cell proliferation. However, in mad numb double mutant clones, there was a reduction in clone size due to loss of ISC in many clones.

A much stronger phenotype was observed when numb mutants were subject to treatment of tissue damaging agents Bleomycin, which causes damage in different ways than DSS. Bleomycin as previously shown to be causing mainly enterocyte damage, and therefore disrupt BMP signaling from ECs more likely. Therefore, this treatment together with loss of numb led to a highly significant reduction of ISC in clones and reduction of clone size/proliferation. One improvement is that it is not clear whether the authors discussed the nature of the two numb mutant alleles used in this study and the comparison to the strength of the RNAi allele. Because the phenotypes are weak and more variable, the use of specific reagents is important.

We have included information about the two numb alleles in the “Materials and Methods”. numb^15^ is a null allele, and the nature of numb^4^ has not been elucidated. According to Domingos, P.M. et al., numb^15^ induced a more severe phenotype than numb^4^ did. Consistently, we also found that more numb^15^ mutant clones were void of stem cell than numb^4^ mutant clones.

Furthermore, the use of possible activating alleles of either or both pathways to test genetic enhancement or synergistic activation will provide strong support for the claims.

Activation of BMP (esgts>Tkv^CA^) alone induced stem cell tumor (Tian et al., 2014) whereas overexpression of Numb did not induce increase stem cell number although overexpression of Numb in wing discs produced phenotypes indictive of inhibition of N (our unpublished observation), making it difficult to test the synergistic effect of activating both BMP and Numb.

**Reviewer #1 (Recommendations for the authors):**
- Cartoon of RGT in Fig 4 needs to be improved. We need to know what chromosome harbors the esgts. It is not sufficient to simply put the location of the ubi-GFP and ubi-RFP (on 19A) and not show the location of other components of the RGT system.

Thank you for the suggestion. We have revised the cartoon in Fig. 4 to include all three pairs of chromosomes and indicate where the esgts driver and UAS-RNAi are located. In addition, we have included the genotypes for all the genetic experiments in the Method section.

- Quantification of the results in Fig. 1

Quantification of Fig.1 has been added.

- The authors need to explain the premise more carefully (see above) and explain whether or not they tested put, numb double knockdowns.

We have explained why not testing put numb double RNAi (see above).

**Reviewer #2 (Recommendations for the authors):**
The number of times the experiments have been performed would be useful to include.

This information has been added in the figure legends.